# Surface and Electrical Characterization of Bilayers Based on BiFeO_3_ and VO_2_

**DOI:** 10.3390/nano12152578

**Published:** 2022-07-27

**Authors:** Jhonatan Martínez, Edgar Mosquera-Vargas, Víctor Fuenzalida, Marcos Flores, Gilberto Bolaños, Jesús Diosa

**Affiliations:** 1Grupo de Transiciones de Fase y Materiales Funcionales, Departamento de Física, FCNE, Universidad del Valle, Santiago de Cali 76001, Colombia; martinez.jhonatan@correounivalle.edu.co; 2Centro de Excelencia en Nuevos Materiales (CENM), Universidad del Valle, Santiago de Cali 76001, Colombia; 3Laboratorio de Superficies y Nanomateriales, Departamento de Física, FCFM, Universidad de Chile, Av. Blanco Encalada 2008, Santiago de Chile 837.0415, Chile; vfuenzal@ing.uchile.cl (V.F.); mflorescarra@ing.uchile.cl (M.F.); 4Grupo de Física de Bajas Temperaturas, Universidad del Cauca, Popayán 190002, Colombia

**Keywords:** thin films, BiFeO_3_/VO_2_, solid–solid interface, surface characterization, electrical property

## Abstract

Thin films of BiFeO_3_, VO_2_, and BiFeO_3_/VO_2_ were grown on SrTiO_3_(100) and Al_2_O_3_(0001) monocrystalline substrates using radio frequency and direct current sputtering techniques. To observe the effect of the coupling between these materials, the surface of the films was characterized by profilometry, atomic force microscopy, and X-ray photoelectron spectroscopy. The heterostructures, monolayers, and bilayers based on BiFeO_3_ and VO_2_ grew with good adhesion and without delamination or signs of incompatibility between the layers. A good granular arrangement and RMS roughness between 1 and 5 nm for the individual layers (VO_2_ and BiFeO_3_) and between 6 and 18 nm for the bilayers (BiFeO_3_/VO_2_) were observed. Their grain size is between 20 nm and 26 nm for the individual layers and between 63 nm and 67 nm for the bilayers. X-ray photoelectron spectroscopy measurements show a higher proportion of V^4+^, Bi^3+^, and Fe^3+^ in the films obtained. The homogeneous ordering, low roughness, and oxidation states on the obtained surface show a good coupling in these films. The I-V curves show ohmic behavior at room temperature and change with increasing temperature. The effect of coupling these materials in a thin film shows the appearance of hysteresis cycles, I-V and R-T, which is typical of materials with high potential in applications, such as resistive memories and solar cells.

## 1. Introduction

Bismuth ferrite (BiFeO_3_ (BFO)) is a multiferroic material exhibiting the coexistence of ferroelectricity (FE) and antiferromagnetism (AF) at room temperature [1,2,3,4,5]. In thin film form, the FE and AF order parameters of multiferroics could be affected for the phenomenon of magnetostriction [1,2,3,4] due to the change in the dimensions of the material as a consequence of stresses generated at the contact interface, either with the substrate or with another material.

Studies of the phenomena that lead to the improvement of magnetostriction and/or ferroelasticity properties focus mainly on changes in the crystalline structure of a material by the incorporation or substitution of other elements [6,7,8]. However, another alternative is the coupling of different materials in thin film heterostructures (see [9,10,11]). Therefore, studying the coupling of the BFO with a material that can generate stress in its crystalline lattice with a structural phase transition is shown as a promising path. Here, the intrinsic capacity of these structures to host the ions of a different chemical nature and size, as well as the coupling of their properties through a contact interface, allows the refining of the physicochemical properties of these compounds in a wide range of possibilities.

To study this mechanism of coupling at the interface with BFO, the vanadium dioxide (VO_2_) system was used, which has been widely analyzed due to the structural change it undergoes from VO_2_ (monoclinic) to VO_2_ (tetragonal) at a temperature of 340 K, showing weak ferromagnetic behavior [12,13]. The combination of these materials in heterostructures gives access to the study of mixed properties and their control.

As far as we can determine, there are currently no reports in the literature that show the growth of BFO thin films over VO_2_. This is mainly due to the incompatibility between the crystalline structures analyzed. However, there are reports on the growth of bilayers with similar crystalline structures or lattice parameters close to those of BFO and VO_2_ that presented good coupling, compatibility, and interesting physical properties, such as the TiO_2_ and de BaTiO_3_. Good compatibility might, therefore, be expected in the bilayer system based on BFO and VO_2_ since the interface effects are predominant and detectable in analogous materials [14]. De la Venta et al. [12,13,15,16] showed the change in the coercivity of thin films of nickel deposited on V_2_O_3_ and VO_2_, a change attributed to the structural phase transition of V_2_O_3_ and VO_2_. Saerbeck et al. [17] studied and compiled information about the coupling between magnetism and the structural phase transition by interfacial tension in different systems of La_0_._7_A_0_._3_MnO_3_ (A = Sr,Ca), Fe_3_O_4_, CoFe_2_O_4_, FeRh, Ni, Fe, and Co, using, as a base, the structural change in the VO_2_, V_2_O_3_, and BaTiO_3_ compounds, resulting in structures with hybrid properties that manage to modify and/or control the magnetic behavior through the effects of tension and inverse magnetostriction. A recent study [18] reports the progress in BFO-based superlattices, which the physical properties of these multifunctional materials have not been explored enough.

On the other hand, Lee et al. [19] reported that BaTiO_3_ presents a perovskite-type crystalline structure like that of BiFeO_3_ with magnetostriction effects. They reported on the modification of the tension of SrRuO_3_ and La_0_._67_Sr_0_._33_MnO_3_ films induced by the phase transition of the BaTiO_3_ substrate, observing the effects of biaxial stress induced on the electric and magnetic transport of these compounds.

Furthermore, Burbure et al. [20] studied the orientation and phase relationship between TiO_2_ films and BaTiO_3_ substrates; considering that TiO_2_ has the structure and lattice parameters of VO_2_, they determined that despite having a high mismatch index, bilayers that present good coupling and epitaxy were grown in different crystalline planes. Feng [21] and Sarkar et al. [22] worked on the development of multifunctional nano-heterostructures of BiFeO_3_/TiO_2_, analyzing photo-ferroelectricity, transport, and non-volatile switching resistances. Zhang et al. [23] used combinatorial substrate epitaxy (CSE) in 150 samples to determine the phase and orientation relationships in the BiFeO_3_/TiO_2_ bilayers. They also proposed the configuration scheme structure whereby the coupling between the layers of these materials with mismatches is presented. However, it was shown that the structural coupling of both the anatase phase of TiO_2_ (like VO_2_ (M)) and the rutile phase (like VO_2_ (R)) is ordered.

In this article, we present a new BFO/VO_2_/Al_2_O_3_ heterostructure with very interesting and promising properties. Also, an analysis of the surface, growth rate, roughness, and grain size were carried out. In addition, an electrical study was carried out where notable changes in the electrical properties of the bilayers were shown when subjected to different temperatures.

## 2. Experimental Details

Thin films of BFO, VO_2_, and BFO/VO_2_ were grown on SrTiO_3_(100) and Al_2_O_3_(0001) substrates using RF and DC sputtering under internal pressure of 4.5 × 10^−4^ mbar and atmospheres of argon and oxygen. All materials, targets (BFO and metallic vanadium, V, with a purity of ≥99.99%), and substrates were obtained from Plasmaterials, Inc. The VO_2_ films were grown in an O_2_-atmosphere at a pressure of 1.72 mbar, power of 53 W, and substrate temperature of 450 °C. In turn, the top BFO layer was deposited in an argon atmosphere (2 × 10^−1^ mbar) at a power of 100 W and a substrate temperature of 550 °C. After deposition, both the single VO_2_ films and the heterostructures were annealed in-situ for 20 min in 20% of O_2_ and 80% of Ar-mixed atmosphere (*p* = 2.4 × 10^−1^ mbar) at the same growth temperature of 550 °C. Monolayers of BFO and VO_2_ with “steps” were obtained, covering a section of the substrate with an alumina mask, allowing the deposition of the material only in one region of the substrate.

Contact profilometry measurements were performed using KLA Tencor D-120 equipment with a measuring range from 1 nm to 100 µm to determine the thickness of the thin films and their growth rate. The surface morphology of the films was examined with an atomic force microscope (AFM, Omicron SPM1), and the Gwyddion software was used to determine the statistical parameters of roughness. The images were processed with an adjusted linear plane to eliminate the inclination before the statistical analysis. The average grain diameter was obtained using ImageJ software. This software was used to delineate the perimeter of the grain, assuming a circular surface. The chemical composition was studied by X-ray photoelectron spectroscopy (XPS) using PHI 1257 spectrometer (PerkinElmer) equipped with a hemispheric electron energy analyzer and an X-ray source with Al Kα radiation (hγ = 1486.6 eV). The XPS spectra were adjusted using the Multipack and XPSpeak programs. The energy scale was calibrated by assigning 284.8 eV to the C1*s* peak corresponding to adventitious carbon. Electrical characterization with resistance curves as a function of temperature R(T) was carried out to study the metal-insulator transition behavior of the VO_2_ compound and of the BFO/VO_2_ system; additional measurements of the current versus the voltage, I–V, were made by the method of Van der Paw or the method of the four points, and finally, the results obtained were correlated.

## 3. Results and Discussion

### 3.1. Profilometry and AFM Studies

Figure 1 and Figure 2 show the profilometry measurements of the VO_2_/Al_2_O_3_ thin films. In Figure 1, the approximate thickness of 77.5 nm provided by the step height parameter (StpHt) indicates a growth rate of 1.3 nm per minute. Instead, in the 3D profile, the uniform height regions can be seen moving away from the step, denoting an ordered growth. In the case of the BFO/SrTiO_3_ thin films (see Figure 3 and Figure 4), the thickness of the BFO films was 106.5 nm, on average, which indicates a growth rate of 2.4 nm per minute under the BFO growth parameters. In these films, the step is more marked, possibly because the mask was much better adhered to and did not allow diffusion between the film and mask. Although the maximum height, Z, is lower than in the VO_2_ films, there is less homogeneity, and there are several zones with different heights.

Surface roughness plays an important role in the effects generated at the interface of the films. For example, in multiferroics systems, the roughness can influence the magnetic and electrical properties of these films. Therefore, the surface of the individual layers and the BFO/VO_2_ heterostructures were analyzed by AFM (see Figure 5, Figure 6 and Figure 7) [24]. Good granular ordering and the RMS roughnesses between 1 and 5 nm were observed for the individual layers and between 6 and 18 nm for the bilayers (see Figure 8). The films exhibited a dense morphology, a continuous surface, and a fine grain microstructure without cracks. The samples with a greater presence of high peaks are due to the agglomerations of the particles that appear in longer growth times, in the same way the grain size for long growth times is smaller and with more uniformity.

Figure 5 shows the image of the surface scan of a section of 1 × 1 μm^2^ and a histogram with the average grain size, which was 26 nm for a monolayer film of VO_2_. In these monolayers, the surface roughness and grain size increase with the deposition time. However, Figure 6 reveals that the grain size of the BFO films is highly homogeneous. No large peaks or valleys were observed. The average grain size is 20 nm, and the roughness does not seem to increase significantly with the increase in thickness.

Figure 7 shows that the grain size of the BFO/VO_2_ films is 63 nm and increases to 67 nm when the BFO layer is 30 and 60 nm, respectively. This could be because the grain geometry is more elongated in the bilayers of 30 nm and roughness of 10.10 nm than with a bilayer of 60 nm and roughness of 16.62 nm. This shows us a more elongated grain geometry in the bilayers, especially the bilayers with greater thicknesses. This elongation in the grains influences the roughness, which increases with respect to the individual layers; this translates into greater growth by the islands and agglomerations of scattered particles with greater height.

Thus, the literature reports that these films grow layer by layer following the structure of the terraces and substrate steps [25,26,27,28,29]. The atoms reaching the surface of the substrate are placed in the middle of the step, and, from there, the structure grows in a 2D form until there is a complete coalescence between the different levels of the layers. The atoms deposited on the substrate are more strongly attached to the next layer and so on. Growth with a larger and not-so-homogeneous grain size can be seen for the top layer of BFO/VO_2_/Al_2_O_3_ with respect to the individual layers [25]. A degree of organization and the absence of cracks are observed, indicating good diffusion and coupling between the BFO, VO_2_, and substrate layers. These characteristics seen in the films contribute to the creation of additional stress that increases with thickness, favoring the formation of islands in the next layer to be deposited. This stress can determine a shift in the critical transition temperature of VO_2_ and, therefore, in the bilayer [26,27,28].

Figure 8 presents a summary of the roughness of the grown films. A semi-exponential upward trend can be observed, showing us how the roughness increases in these films as the thickness and number of layers of these two thin-film materials increase.

### 3.2. Chemical States

The chemical studies were carried out through XPS analysis to detect the chemical elements present on the surface of these films. The analysis of the surfaces by XPS is a very valuable resource when determining both quantitatively and qualitatively the composition and chemical state of the surfaces. However, the analysis by this technique of first-row transition metals and their oxides and hydroxyls is a challenge due to the complexity of their 2*p* spectra, resulting from asymmetric peaks, the complex division of multiplets, and the uncertain and overlapping union of energies [30].

The XPS measurements are presented for four samples, two monolayers (VO_2_/Al_2_O_3_ and BFO/SrTiO_3_), and two bilayers (BFO/VO_2_/Al_2_O_3_, with BFO layers of 30 and 60 nm), respectively. These measurements were made in a broad spectrum, and the narrow scans are shown in Figure 9, Figure 10, Figure 11 and Figure 12. All spectra were adjusted with mixed Lorentzian and Gaussian curves. The Mo signal is from the sample holder.

Figure 9 shows the spectra of the VO_2_/Al_2_O_3_ monolayer in the blue color. In Figure 9a, the absence of impurity peaks, except for molybdenum (Mo) coming from the sample holder used, can be seen. However, in the narrow scan spectrum (Figure 9b), the respective curve adjustment was obtained for the V^+4^ peaks (516.6 eV and 524.24 eV), V^+5^ peaks (517.94 eV and 525.58 eV), O_L_ peak (530.84 eV), and O_H_ peak (532.81 eV), respectively. Here, the O_L_ and O_H_ peaks correspond to the bonds of oxygen with hydrogen and with the crystal lattice of the compound studied and indicate that the films are very susceptible to oxidation in the environment. In addition, a spin-orbit splitting of 7.64 eV typical of vanadium oxides was observed, and it is evident that the predominant oxidation state is V^+4^, which corresponds to VO_2_. On the other hand, Figure 9a shows the spectra of the BFO/SrTiO_3_ monolayer in the green color. Signals from bismuth, iron, oxygen, and molybdenum (from the sample holder) were observed. However, using a narrow scan is possible to determine the composition of the thin film (Figure 10a and Figure 11a). Signals from Fe(2*p*) doublet (at 710.02 and 723.62 eV) and Bi(4*f*) doublet (at 156.84 and 162.15 eV), including metallic Bi, were observed. The binding energies’ differences for each are 13.6 eV and 5.31 eV, respectively, which constitutes a response of iron oxides and bismuth oxides. On the other hand, the high-resolution spectra of oxygen (O1*s*, see Figure 12a) show the presence of the oxides of the lattice (metallic oxides, O_L_) and hydroxyl oxygen (O_H_) in the thin films. Therefore, it is clearly observed that the predominant oxidation states are Fe^3+^ and Bi^3+^, corresponding to the structure of BiFeO_3_.

For the bilayer thin films, see Figure 9a in the red and black colors; a BFO of 30 nm can be seen in Figure 10, Figure 11 and Figure 12b, and a BFO of 60 nm can be seen in Figure 10, Figure 11 and Figure 12c. We can observe that the Fe(2*p*) doublet and Bi(4*f*) doublet’s binding energies difference of 13.6 eV and 5.31 eV are maintained. Therefore, the BFO thin films with a layer of 30 nm exhibit more oxidation states, as well as a shift towards higher binding energies. However, an ion bombardment revealed that the amount of Fe^3+^ increases in the BFO thin films, unlike the surface where there was a higher presence of the Fe^2+^ oxidation state. In the bilayers, there was a small displacement of the maximums towards higher values of binding energy; however, this did not increase the number of oxidation states, and it can be noted that at lesser layers, Fe^3+^ increases in proportion. On the other hand, it is evident that Bi^3+^ is very predominant in both the monolayers and bilayers. Instead, there is an increase in the contribution of the oxides of the lattice (metallic oxides, O_L_) for the bilayers, which is a good indication that the surface area is more homogeneous and that there is less contamination. In addition, hydroxyl oxygen (O_H_) increases in the thin films with a lower growth power and in the eroded sample, which may suggest that these thin films could be more photocatalytic.

Figure 13 and Table 1 show a summary of the atomic percentages of the thin films analyzed in this work. It is observed that the VO_2_ films most indicated to be coupled with the BFO films are those with 60 min of growth since they have a higher percentage of vanadium on the surface and present a higher proportion of V^4+^. On the other hand, the most suitable BFO films to couple in the bilayer with VO_2_ are those with a BFO layer of 30 nm because they show iron with a higher atomic percentage in the surface area and a higher Fe^3+^ ratio.

Finally, it should be noted that growing this type of film in a bilayer without previous antecedents constituted a challenge, not only because of the coupling of their crystalline structures before and after the VO_2_ transition temperature but also because the upper layer (BFO) is grown at a higher temperature than the lower VO_2_ layer and even its annealing temperature at 470 °C. This has, therefore, a deoxidation effect on the VO_2_ film, turning it metallic and losing its metal-insulator transitions (MIT), which is not desirable, since coupling with a material that presents a structural phase transition is essential to generate a new line of applications and studies of the physical properties of these bilayers. Therefore, the VO_2_ films were carefully over-oxidized, anticipating these losses when the upper BFO film grew, and in this way, a growth methodology of these bilayers was obtained.

### 3.3. Electrical Characterization

One of the best indications of good coupling in the bilayers based on BFO and VO_2_ is that these also present hysteresis curves with a decrease of several orders of magnitude of electrical resistance in a fairly short time (see Figure 14 left); even so, there is a leftward shift in the transition temperature with respect to the 340 K of VO_2_. This lower value (around ~325 K) can be caused by an increase in the interfacial tension in VO_2_ since there are studies, such as those reported in [26,31], which show us that there is a relationship between the increase in the tension of the crystal lattice in VO_2_ and the decrease in critical temperature. This result corroborates the XRD analysis that shows tensions in its crystal lattice; this would be not only with the substrate of Al_2_O_3_ but also with the film on it (BFO). Additionally, this possibly contributes to an increase of the voltage of the lattice in the VO_2_ and, therefore, to decrease the transition temperature.

The electrical properties of the BFO/VO_2_/Al_2_O_3_ heterostructures are shown in Figure 14. A well-defined MIT with a resistance change of two and three orders of magnitude (ΔR~100 and ~1000) can be seen in (a) and (b) with a T_MIT_ between ~322 and 327 K (hysteresis width between ΔH ≈ 4 and 3 K) for the 30 and 60 nm bilayers, respectively. T_MIT_ is estimated by the well-known expression T_MIT_ = ½(T_c_ – T_h_). Here, (T_c_ – T_h_) represents the transition temperatures at the center of the derived curve (d[log(R)]/dT) during the cooling and heating procedure, respectively (inset in Figure 14 on the right). The observation of an MIT, which improves the conductivity of the studied system, confirms the presence of the VO_2_(M) phase in the lower film. Here, it should be noted that the VO_2_(B) phase exhibits neither MIT nor thermal hysteresis phenomena [32]. The resistance of VO_2_(B) decreases exponentially with increasing temperature [33]. It is striking that the measured T_MIT_ of the polycrystalline film VO_2_ in the heterostructure is relatively low compared to that reported for individual crystals or epitaxial thin films [34]. This finding, along with the observed resistance change of several orders of magnitude, is in agreement with the results reported for polycrystalline VO_2_ thin films [35,36].

From Figure 14, we can conclude that the thermal stability of the coupled material strongly depends on the transition temperatures observed for each bilayer; however, by having reproducibility, we can see the heterostructure as a single coupled material with an operating range between 300 K and 380 K; within this range, a very interesting behavior is observed that could give rise to the study and/or application of the control of electrical, ferroelectric, or magnetic properties.

It is worth mentioning that the growth of polycrystalline thin films of VO_2_ with reduced T_MIT_ is technologically interesting and constitutes a current research topic in solid state physics [37]. Although the T_MIT_ of the polycrystalline thin film VO_2_ can be effectively adjusted and various approaches have been adopted to explain the origin of the reduced T_MIT_, there is no consensus on this matter. Therefore, the true physical mechanism behind the reduction in the T_MIT_ value of polycrystalline VO_2_ thin films remains an open question.

Although understanding the phase transition in thin-film VO_2_ is challenging, it is probably related to a network distortion process. In a simplified image, it has been argued that a charge density wave along the c-axis of rutile [38], with the wave vector 2cR, forms during the transition. As a result, the unit cell along the c-axis is duplicated, generating a periodic distortion of the network in this direction. A slight rotation of the dimers with respect to the c-axis has also been verified [38]. As for the phase transition of VO_2_ polycrystalline films, it is evident that the phase transition, in this case, is easier than that of VO_2_ monocrystals [27]. The main feature of VO_2_ polycrystalline films is the deformation compatibility between differently oriented grains. Therefore, it is to be expected that the grains oriented along specific directions may have irregular distortions of the network.

To finish with the characterizations carried out, it was decided to study I-V curves to analyze the electrical behavior of the bilayer thin film with a BFO layer of 30 nm as a first approximation to future studies of these bilayers in functional applications for storage devices, such as memresistors. The bilayer thin film with a BFO layer of 30 nm was chosen because it was the one that, in percentage, varied its electrical and magnetic responses more compared to the MIT of VO_2_ and, therefore, the film that has the greatest potential to work with nanotechnological applications or prototypes.

Figure 15 shows the I–V curves of BFO/Al_2_O_3_ (a BFO layer of 30 nm) with varying voltage from 0 V to 200 V at different temperatures (Figure 15 (up)) and from −200 V to 200 V at a temperature of 320 K (Figure 15 (down)). Here, it is evident how the film progressively goes from having an ohmic behavior at room temperature to having a hysteresis cycle typical of resistive memories, which has a maximum width of ~20 V and, as the temperature decreases, this hysteresis cycle closes.

Studying more thoroughly the I-V of the bilayer at 320 K, we can observe a difference in the symmetry of the resistive cycle with positive and negative voltages; however, the negative voltage cycle is more closed, an effect that agrees with what is reported in [39,40]. Here, it is denoted that it is typical behavior of these memresistor materials that they change their response with each measurement.

On the other hand, effects such as magnetoelectric coupling and its behavior at different fields and with temperatures before and after the transition would be the next step for the study of these heterostructures with a high mismatch and FE/AF/MIT coupling.

## 4. Conclusions

It was possible to obtain the growth parameters for films based on BFO and VO_2_, together with a methodology that allows obtaining a good coupling between films and the desired oxidation states of vanadium, bismuth, and iron. BFO and VO_2_ monolayers were obtained with an ordered growth, good adherence, roughness (between 0.2 and 2.3 nm), and grain size (between 20 and 32 nm). In addition, an ordered growth without cracks is evidenced. AFM measurements show differences in the topography of the individual layers and bilayers, a change in the average grain diameter from 20 nm to 67 nm, and a change in the roughness from 0.2 nm to almost 16 nm in the BFO/VO_2_ bilayers.

The AFM measurements show differences in the topography of the individual monolayers and bilayers; a change in the average grain diameter from 20 nm to 67 nm was observed, and a change in roughness from 0.2 nm to nearly 16 nm was observed in the BFO/VO_2_ bilayers. In addition, an orderly growth without cracks (AFM) is evident. The homogeneity observed in the films with the best results will possibly contribute to obtaining good results in characterizations of the current, voltage, resistance, and behavior against the electric and magnetic fields. It was observed by means of XPS that the predominant oxidation state in the individual and bilayer thin films is VO_2_. However, there is also a presence in the films of the V_2_O_5_ phase. The BFO and BFO/VO_2_/Al_2_O_3_ films show the presented BiFeO_3_ as the predominant phase.

It is pertinent to say that these new heterostructures are an excellent example of how synergistic effects in thin film physics can help drive new studies in already known areas and materials.

Furthermore, thin films show a structural phase change with a resistance change of several orders of magnitude, both for the monolayers and for the bilayers. Also, hysteresis loops are presented here, which is an excellent indication to work these films in memresistor and solar cells.

## Figures and Tables

**Figure 1 nanomaterials-12-02578-f001:**
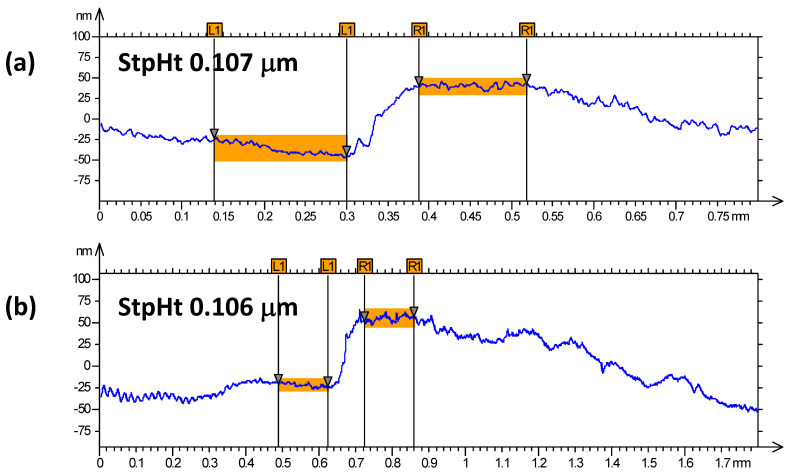
(**a**,**b**) Profilometry measurements at two different points of thin films of VO_2_ on Al_2_O_3_.

**Figure 2 nanomaterials-12-02578-f002:**
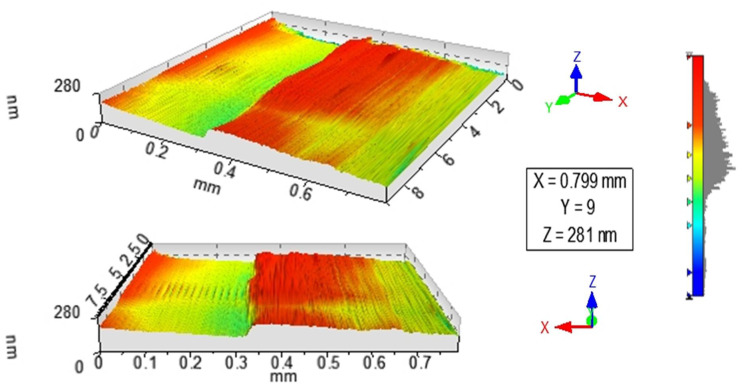
3D image obtained by profilometry of the surface of thin films of VO_2_ on Al_2_O_3_.

**Figure 3 nanomaterials-12-02578-f003:**
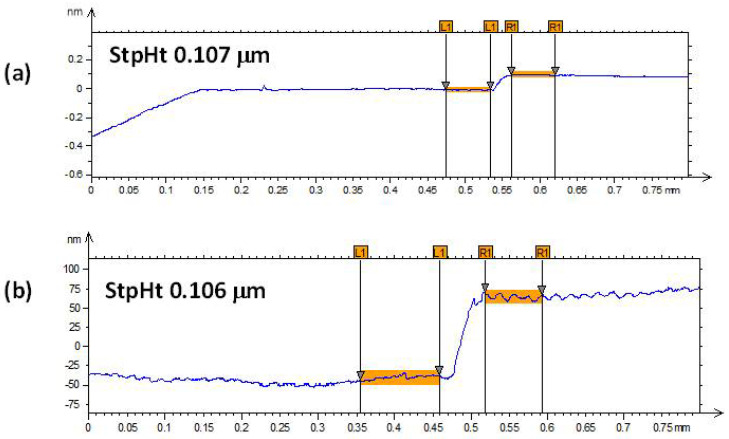
(**a**,**b**) Profilometry measurements at two different points of thin films of BFO on SrTiO_3_.

**Figure 4 nanomaterials-12-02578-f004:**
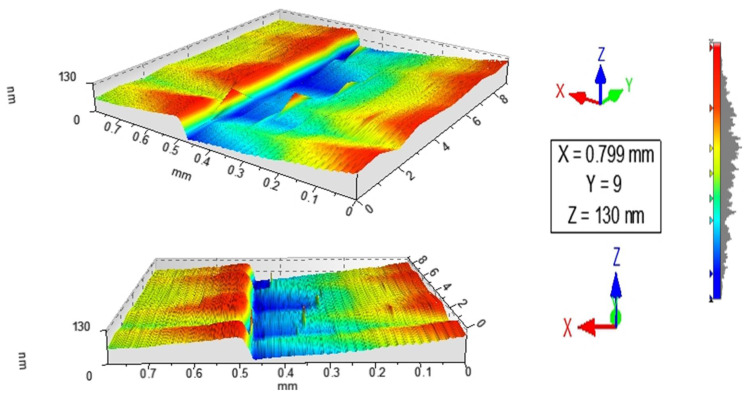
3D image obtained by profilometry of the surface of thin films of BFO on SrTiO_3_.

**Figure 5 nanomaterials-12-02578-f005:**
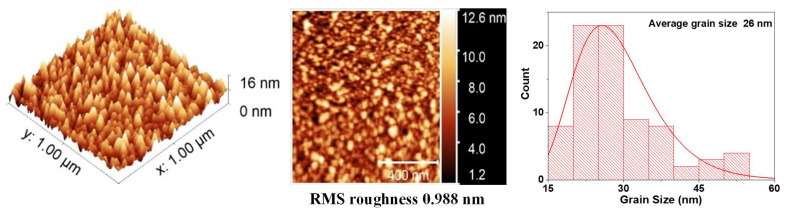
AFM measurements and average grain size for VO_2_ thin films.

**Figure 6 nanomaterials-12-02578-f006:**
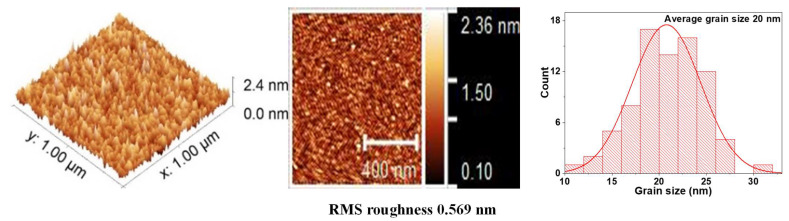
AFM measurements and average grain size for BFO thin films.

**Figure 7 nanomaterials-12-02578-f007:**
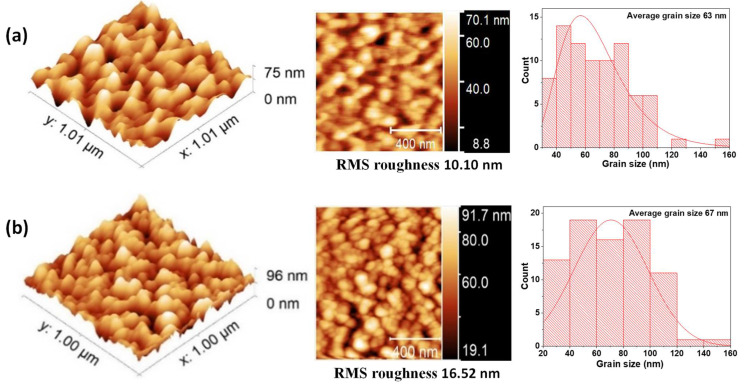
AFM measurements and average grain size for BFO/VO_2_ thin films with BFO layers of (**a**) 30 nm and (**b**) 60 nm.

**Figure 8 nanomaterials-12-02578-f008:**
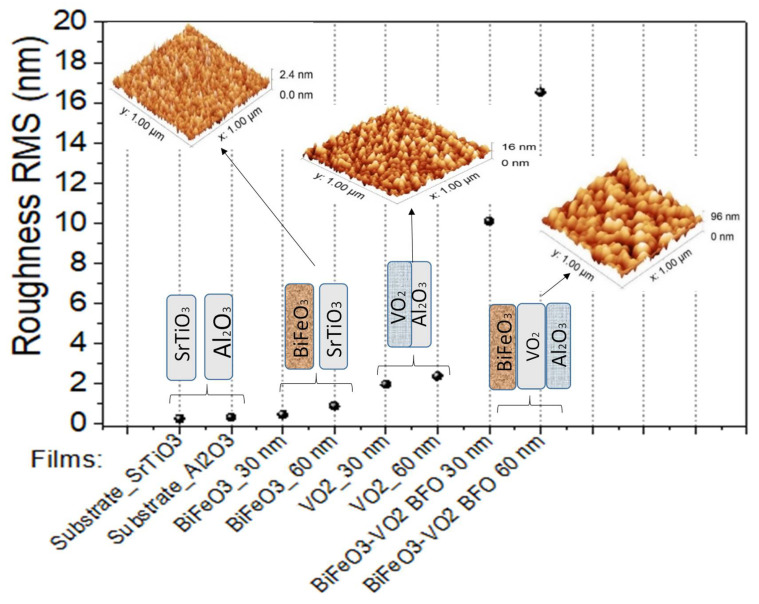
Roughness of films based on BFO and VO_2_.

**Figure 9 nanomaterials-12-02578-f009:**
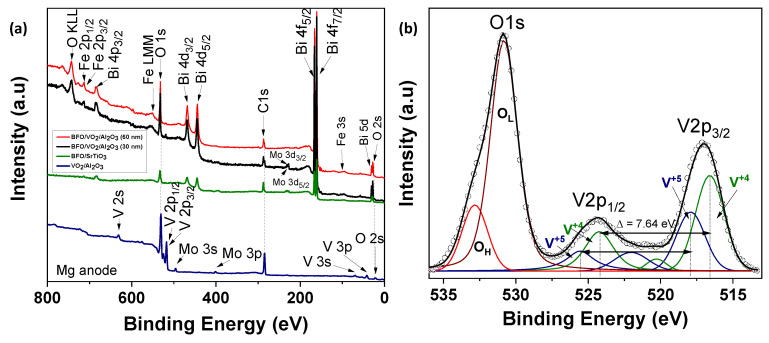
(**a**) The survey scans of the XPS spectra for the monolayers (BFO and VO_2_) and bilayers (BFO/VO_2_/Al_2_O_3_ with BFO 30 nm and 60 nm) (**b**) narrow scan XPS spectrum of VO_2_ monolayer.

**Figure 10 nanomaterials-12-02578-f010:**
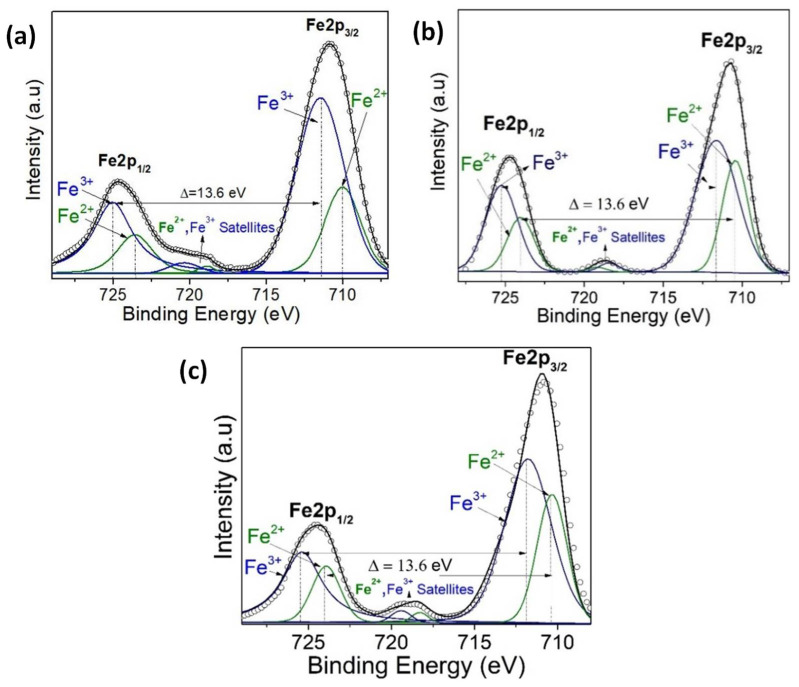
(**a**) XPS narrow scan of Fe 2p signals for (**a**) BFO/SrTiO_3_, (**b**) BFO/VO_2_/Al_2_O_3_ with a BFO of 30 nm, and (**c**) BFO/VO_2_/Al_2_O_3_ with a BFO of 60 nm.

**Figure 11 nanomaterials-12-02578-f011:**
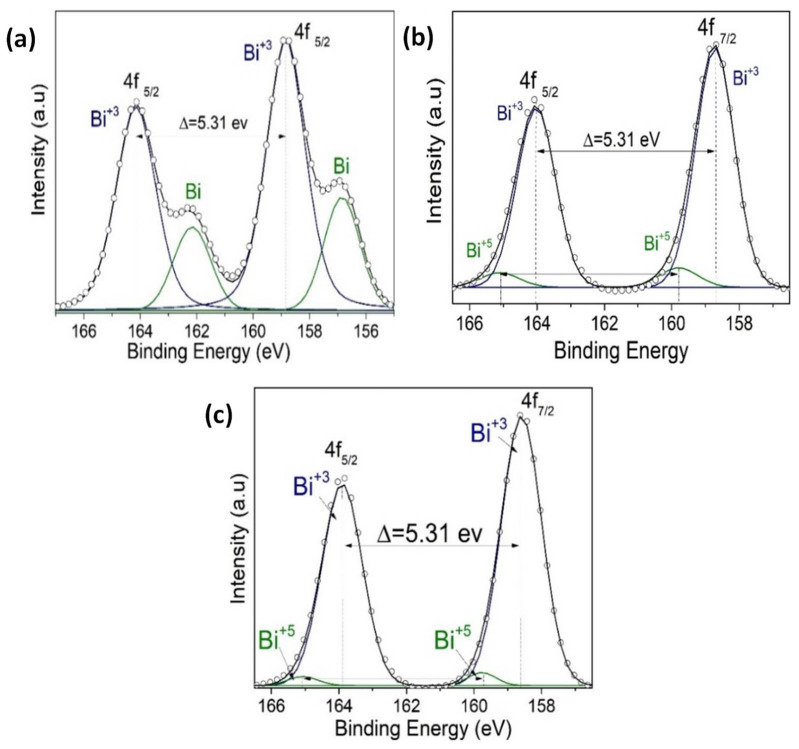
(**a**) An XPS narrow scan of Bi 4f signals for (**a**) BFO/SrTiO_3_, (**b**) BFO/VO_2_/Al_2_O_3_ with a BFO of 30 nm, and (**c**) BFO/VO_2_/Al_2_O_3_ with a BFO of 60 nm.

**Figure 12 nanomaterials-12-02578-f012:**
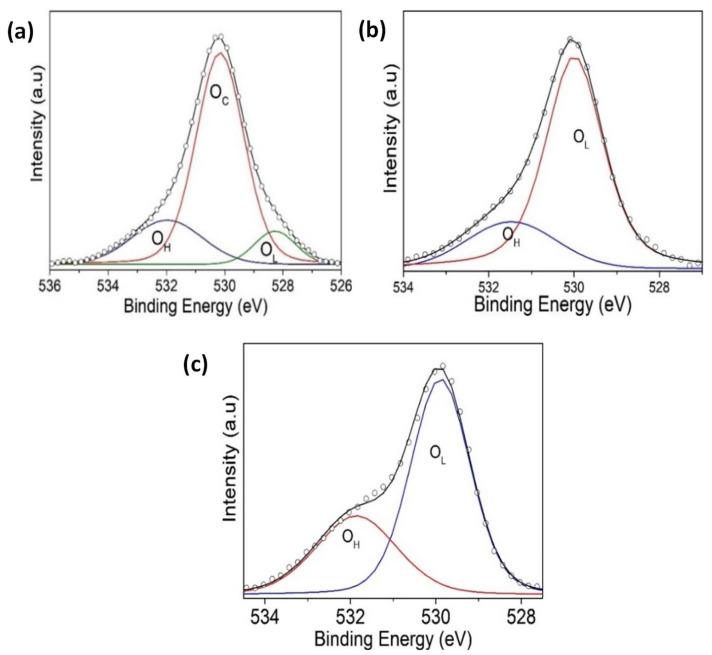
An XPS narrow scan of O 1s signals for (**a**) BFO/SrTiO_3_, (**b**) BFO/VO_2_/Al_2_O_3_ with a BFO of 30 nm, and (**c**) BFO/VO_2_/Al_2_O_3_ with a BFO of 60 nm.

**Figure 13 nanomaterials-12-02578-f013:**
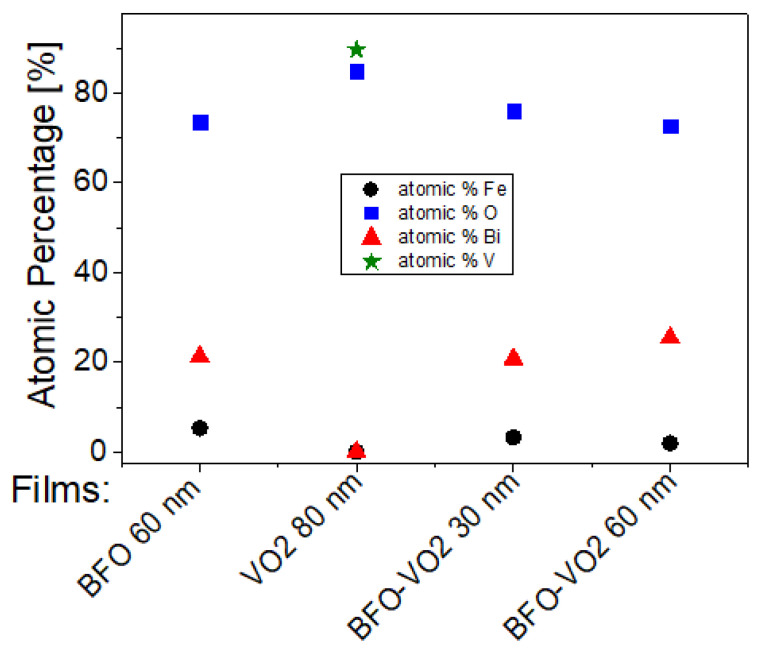
Atomic percentages on the surface of the selected films.

**Figure 14 nanomaterials-12-02578-f014:**
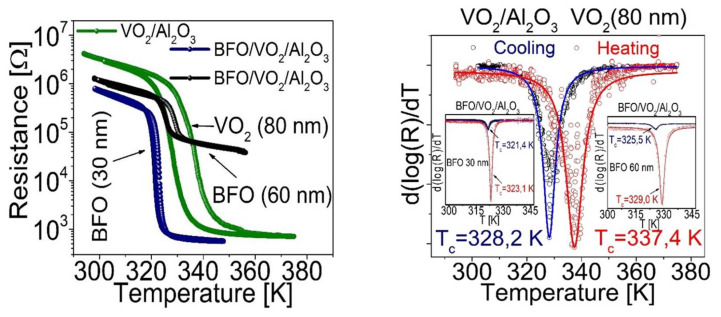
(**left**) Resistance curves as a function of temperature for VO_2_/Al_2_O_3_ (VO_2_ 80 nm), BFO/VO_2_/Al_2_O_3_ (BFO layer of 30 nm) and BFO/VO_2_/Al_2_O_3_ (BFO layer of 60 nm) (**right**) derived from electrical resistance as a function of temperature for the films obtained.

**Figure 15 nanomaterials-12-02578-f015:**
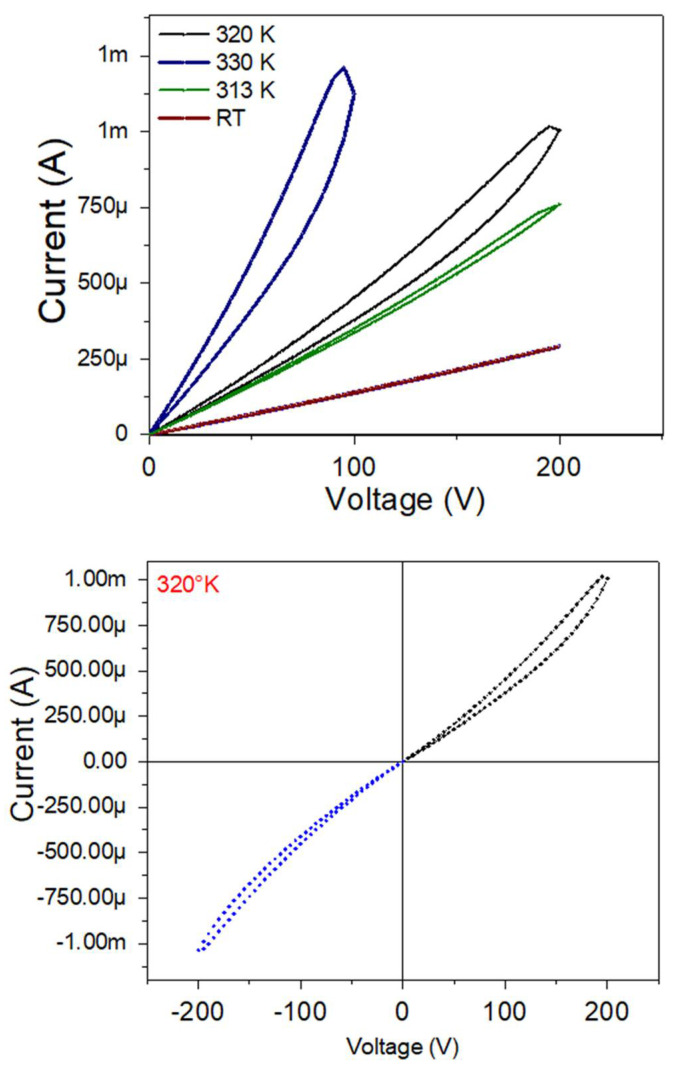
I-V curves for a BFO/VO_2_/Al_2_O_3_ bilayer, with a BFO layer of 30 nm with different temperatures (**up**) and at 320 K with positive and negative voltage values (**down**).

**Table 1 nanomaterials-12-02578-t001:** Summary of atomic percentages on the surface of the thin films obtained.

Films	Atomic% Fe	Atomic% O	Atomic% Bi	Atomic% C	Atomic% V
BFO 60 nm	5.28	73.46	21.26	59.49	--
VO_2_ 80 nm	--	84.91	--	79.66	89.61
BFO-VO_2_ 30 nm	3.25	75.98	20.77	52.24	--
BFO-VO_2_ 60 nm	1.92	72.64	25.44	42.26	--

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
