# Peer review of "Surface and Electrical Characterization of Bilayers Based on BiFeO3 and VO2"

_nanomaterials, 2022, doi:10.3390/nano12152578_

Round 1

Reviewer 1 Report

This manuscript offers a concise yet compelling report on a novel nanostructured thin film system consisting in Bi-layers based on BiFeO3 and VO2. The emphasis of the study falls on the impact of the effect(s) of coupling these two compounds in a synergetic thin film exhibiting hysteresis cycles I-V and R-T which indeed is highly interesting from point of view of applications. A very well-planed experimental set-up is employed and masterfully chosen/adapted for the purpose of this task. The really nice and adequate comparative context of characterization (profilometry, AFM and XPS) results throughout the study strongly contributes to reliable scrutinization and prompting development of such unusual synergetic material system taking advantage of the synergy effects of coupling two compounds and exhibiting highly useful properties with direct perspectives for applications.

From practical point of view, the reported results thus bring new knowledge and certainly represent an original contribution in the present context.

The authors chose an adequate structure of the manuscript – an excellent point of departure for such a study. Finally, the authors provided a balanced realistic and nicely illustrated presentation of their results and corresponding analysis that is of much scientific and practical interest and adds new knowledge to the field.

In my opinion, the fine detailing in the present work, the insightful and balanced discussion of the results, as well as the excellent, intuitively perceived figures, permit wide circle of readers to utilize the manuscript as a guidance for their potential future work in the same research field. Consequently, this manuscript presents an efficient and beneficial basis for promoting and solving next step challenges in this field.

The manuscript also benefits from a clear motivation, and it is an easy and informative read.

The present manuscript is a significant contribution, this work once published would be quite useful as well as instructive and suggestive in terms of further studies and to a wider readership.

There are some minor issues with this already excellent manuscript that will need to be addressed before becoming suitable for publication, i.e., it can be considered for publication after a minor revision:

1: In the introduction, the authors partly miss some theoretical works whereby aspects of structural features in complex systems, including bismuth ones and with synergistic aspects have been greatly helped, directed, predicted and supported. Such theoretical examples in direct corroboration with synthesis of complex inherently nanostructured systems include: Nanoscale 12 (2020) 19470-19476; Journal of Physics: Condensed Matter 27 (2015) 485306. Such works should be referred to, showing credibility of designing protective/nanostructured coatings concept by theoretical support.

2: The authors should elaborate more on the thermal stability of the coupled material system stating more explicitly range of thermal stability which may be perceived in the present text ambiguously by the reader, since temperature has been discussed mostly in the context of electrical/magnetic properties. This is fully correct, but thermal stability in general should be important too.

3: Did the authors consider, or are they aware of any detailed study of the present system in regarding domain structure versus polarization aspects?

4: Any bonding particulars that deserve more detailed discussion to improve reader’s understanding of the coupled system?

5: It would be useful to mention in the conclusions that these films are an excellent example of synergetic effects in thin film physics.

6: Spell-check and stylistic revision of the paper are still necessary. Some, long sentences, misspellings, etc., still are noticeable throughout the text.

Author Response

Reply to the referee’s comments on the manuscript “Surface and Electrical Characterization of Bilayers Based On BiFeO3 and VO2” (nanomaterials-1817285)

Dear Editor

We would like to express our gratitude to the referees for their enlightening and constructive comments on our paper “Surface and Electrical Characterization of Bilayers Based on BiFeO3 and VO2(nanomaterials-1817285). We also want to recognize the valuable work of the editor and the editorial board alike for their professional management of our manuscript. We look forward to providing a revised version of the paper with the changes suggested by the reviewers. Herein we wish to address the reviewers’ comments and remarks through a point-by-point response in this document. In addition, the revised version of the manuscript, including the highlighted changes, is uploaded. The changes made are summarized below:

Reviewer #1.

1) In the introduction, the authors partly miss some theoretical works whereby aspects of structural features in complex systems, including bismuth ones and with synergistic aspects have been greatly helped, directed, predicted and supported. Such theoretical examples in direct corroboration with synthesis of complex inherently nanostructured systems include: Nanoscale 12 (2020) 19470-19476; Journal of Physics: Condensed Matter 27 (2015) 485306. Such works should be referred to, showing credibility of designing protective/nanostructured coatings concept by theoretical support

Answer: The referee is right. Structures based on the combination and coupling of different compounds are certainly attractive for various practical applications. Following the referee's suggestion, we have added a short paragraph highlighting the importance of these materials both from the point of view of basic knowledge and possible technological application. The references suggested by the referee have been included in the revised version of the manuscript. The following paragraph has also been added:

Studies of the phenomena that lead to the improvement of magnetostriction and/or ferroelasticity properties focus mainly on changes in the crystalline structure of a material by incorporation or substitution of other elements [6]–[8]. However, another alternative is the coupling of different materials in thin film heterostructures, see Refs [9]–[11]. Therefore, studying the coupling of the BFO with a material that can generate stress in its crystalline lattice with a structural phase transition is shown as a promising path. Here, the intrinsic capacity of these structures to host ions of different chemical nature and size, as well as the coupling of their properties through a contact interface, allows refining the physicochemical properties of these compounds in a wide range of possibilities.

2) The authors should elaborate more on the thermal stability of the coupled material system stating more explicitly range of thermal stability which may be perceived in the present text ambiguously by the reader, since temperature has been discussed mostly in the context of electrical/magnetic properties. This is fully correct, but thermal stability in general should be important too.

Answer: Thank you very much for your suggestion, regarding this, we have talked a little more in the following paragraph included in the manuscript.

From Figure 14, we can conclude that the thermal stability of the coupled material strongly depends on the transition temperatures observed for each bilayer, however, by having reproducibility, we can see the heterostructure as a single coupled material with an operating range between 300 K and 380 K, within this range a very interesting behavior is observed that could give rise to the study and/or application of the control of electrical, ferroelectric, or magnetic properties.

3) Did the authors consider, or are they aware of any detailed study of the present system in regarding domain structure versus polarization aspects?

Answer: We are currently not aware of any detailed study of the current system with respect to domain structure versus polarization or magnetization aspects, however, we are in the process of conducting a detailed study to share with the scientific community.

4) Any bonding particulars that deserve more detailed discussion to improve reader’s understanding of the coupled system?

Answer: Thank you for your comment, in reality, there are many important details that require more detailed discussions, however, in order not to make the text longer and make it more enjoyable for the reader, we consider this article and the next one (which we plan to discuss in a few months) are a good basis to arouse the interest of readers, the most promising of this link is the modification of the properties of the BFO by coupling with the VO2 obtaining different electrical responses before and after the insulating metal transition temperature of the VO2.

5) It would be useful to mention in the conclusions that these films are an excellent example of synergetic effects in thin film physics?

Answer: On attending to the referee’s suggestion, we have included a pertinent conclusion: It is pertinent to say that these new heterostructures are an excellent example of how synergistic effects in thin film physics can help drive new studies in already known areas and materials.

6) Spell-check and stylistic revision of the paper are still necessary. Some, long sentences, misspellings, etc., still are noticeable throughout the text.

Answer: Thanks, we did our best to improve the style and long sentences in our article.

The revised version of the manuscript concerning both grammar and style has been corrected by a professional English editing service

Reviewer 2 Report

The manuscript titled “SURFACE AND ELECTRICAL CHARACTERIZATION OF BILAYERS BASED ON BiFeO3 AND VO2” presented by Jhonatan Martínez, Edgar Mosquera-Vargas , Victor Fuenzalida, Marcos Flores, Gilberto Bolaños, Jesús Diosa deals with the characterization of BiFeO3 and VO2 films. However, there are major issues that could be clarified:

  • The abstract should not content the abbreviations, for example, RF, DC, XPS, etc.
  • It does not clear which samples are described in the sentence “The heterostructures have roughnesses between 0.2 and 16 nm and a grain size between 20 nm and 67 nm.”
  • What “Fe region” means?
  • The abstract should be re-written to clearly present the study.
  • The aim of the study should be clarified, it should sound more scientific in global scale. Now the measurements of some physical parameters present like an aim of the study.
  • The description of XPS setup is incorrect and not contemporary. Not lamb, but X-ray gun, the photon energy should be presented in eV, not spectroscope, but setup or spectrometer. Not hγ, but hν. “An analytical aperture 0.8 m in diameter” is incredible, please, check.
  • Figures 1 and 3 present the TIR, Slope and Avg parameters that are not discussed in the manuscript. Could authors explain how the height of the step could evaluate the thickness of the film? Could the authors present the data of ellipsometry? Some measured parameters are presented with high precision, for example, the rate of film growth is 13 Angstrom (!) per minute, that looks unbelievable. It seems that the thickness of film is incorrect measured. Please, provide the measurement error for all measured parameters.
  • The description of XPS study has some incorrect terms, for example, “the chemical states in these thin films”. The “distance between the vanadium doublet of 7.64 eV” is named spin-orbit splitting. The survey spectrum has Mo signal, it looks that authors did not make enough efforts to adjust the position of sample. The size of the films allows fine position adjustment. Mo could be also admixed from synthesize, please, redone XPS measurements, at least for one sample. I recommend authors to combine the similar spectra to one figure, for example, all Fe 2p spectra, etc. The author did not used the correct terms for description of XPS data, in the current form it looks oversimplified. Please, provide the scientific description of XPS spectra. The data presented on figure 13 should be presented in the table.
  • I am not specialist in the electrical characterization, hope, the other reviewers check this part carefully.

The major comment concerns the aim of the study, authors should clarify it. Some additional experiments are demanded to support the author’s ideas (ellipsometry). The XPS study should be re-written in the classic way. The comparison of films under study with the systems synthesized for similar purpose is necessary.

Author Response

Reply to the referee’s comments on the manuscript “Surface and Electrical Characterization of Bilayers Based On BiFeO3 and VO2” (nanomaterials-1817285)

Dear Editor

We would like to express our gratitude to the referees for their enlightening and constructive comments on our paper “Surface and Electrical Characterization of Bilayers Based on BiFeO3 and VO2(nanomaterials-1817285). We also want to recognize the valuable work of the editor and the editorial board alike for their professional management of our manuscript. We look forward to providing a revised version of the paper with the changes suggested by the reviewers. Herein we wish to address the reviewers’ comments and remarks through a point-by-point response in this document. In addition, the revised version of the manuscript, including the highlighted changes, is uploaded. The changes made are summarized below:

Reviewer #2:

1) The abstract should not content the abbreviations, for example, RF, DC, XPS, etc.

Answer: Thank you for your feedback, we have made an improvement to the summary.

2) It does not clear which samples are described in the sentence “The heterostructures have roughnesses between 0.2 and 16 nm and a grain size between 20 nm and 67 nm.”

Answer: You are right, we have corrected the paragraph clarifying the idea.

The heterostructures, monolayer and bilayer based on BiFeO3 and VO2 grew with good adhesion and without delamination or signs of incompatibility between the layers, have roughness between 0.2 and 16 nm and a grain size between 20 nm and 67 nm.

3) What “Fe region” means?

Answer: thanks for the correction, the scientific wording of the text has been improved, in this case, "Fe region", referred to the Fe 2p region between the band energies 700 eV and 730 eV.

X-ray photoelectron spectroscopy measurements show a higher proportion of the V4+, Bi3+ and Fe3+ in the films obtained.

4) The abstract should be re-written to clearly present the study

Answer: We thank the referee for this comment. On attending her/his suggestion we have shorten the abstract in a way that this clearly contains the key point found in the present investigation. The abstract also contains some qualitative and quantitative results.

5) The aim of the study should be clarified; it should sound more scientific in global scale. Now the measurements of some physical parameters present like an aim of the study.

Answer: Many thanks for this suggestion,

In this article we present a new BFO/VO2/Al2O3 heterostructure with very interesting and promising properties, an analysis of surface, growth rate, roughness and grain size is carried out, in addition, an electrical study was carried out where notable changes in the electrical properties of bilayers when subjected to different temperatures.

6) The description of XPS setup is incorrect and not contemporary. Not lamb, but X-ray gun, the photon energy should be presented in eV, not spectroscope, but setup or spectrometer. Not hγ, but hν. “An analytical aperture 0.8 m in diameter” is incredible, please, check.

Answer: Thanks for the comments, we have corrected the description of XPS appropriately and only the relevant data is shown, additionally an email was sent to the laboratory where the measurements were taken in order to corroborate the analytical opening diameter, but this email unfortunately has not yet has been answered.

7) Figures 1 and 3 present the TIR, Slope and Avg parameters that are not discussed in the manuscript. Could authors explain how the height of the step could evaluate the thickness of the film? Could the authors present the data of ellipsometry? Some measured parameters are presented with high precision, for example, the rate of film growth is 13 Angstrom (!) per minute, that looks unbelievable. It seems that the thickness of film is incorrect measured. Please, provide the measurement error for all measured parameters.

Answer: Thank you for your appreciation, you are right, because these values ​​are not part of the main axis of our research, they were removed from the figures, leaving only the relevant data StpHt (step height parameter: is defined as the difference between the average Z value of the left measurement cursor and the average Z value of the right measurement cursor) This value is useful to us as it allows us to calculate the growth rates of the measured thin films. The other values ​​mentioned above are: TIR (Total Indicator Runout: is defined as the difference between the highest and lowest points in the profile, measured between the left and right measurement cursors. This is the same as the maximum peak-to-valley Z deviation), AVR (Average Height: is defined as the arithmetic mean of the Z data between the left and right measurement cursors) and Slope (is based on the ratio of the difference in vertical positions to the difference in horizontal positions of the measurement cursors The slope is reported in degrees, with respect to the horizontal).

8) The description of XPS study has some incorrect terms, for example, “the chemical states in these thin films”. The “distance between the vanadium doublet of 7.64 eV” is named spin-orbit splitting. The survey spectrum has Mo signal, it looks that authors did not make enough efforts to adjust the position of sample. The size of the films allows fine position adjustment. Mo could be also admixed from synthesize, please, redone XPS measurements, at least for one sample. I recommend authors to combine the similar spectra to one figure, for example, all Fe 2p spectra, etc. The author did not used the correct terms for description of XPS data, in the current form it looks oversimplified. Please, provide the scientific description of XPS spectra. The data presented on figure 13 should be presented in the table.

Answer: We have corrected the erroneous terms and we apologize for the Mo signal in the Survey spectrum, this is mainly due to the fact that the samples obtained were grown on very small substrates of 5x5 mm and the sample holder was a Molybdenum cylinder of 3 cm in diameter. , this made it extremely difficult to obtain curves without Mo signal, however we clarify that during the synthesis process a Sputtering chamber was used that was rigorously cleaned and degassed, in which Mo has never been introduced, additionally during the growth of our films, inside the chamber there were only the BiFeO3 and VO2 targets, this tells us that there is contamination in the sintering of the material, finally, the Mo signal is very weak and does not influence the analysis of the elements of interest in this research, as seen in Figure 12 where it was possible to cancel the Mo signal for the measurements in the BFO/VO2/Al2O3 bilayer, see Figure and the table.

Films

Atomic % Fe

Atomic % O

Atomic % Bi

Atomic % C

Atomic % V

BFO 60 nm

5,28

73,46

21,26

59,49

--

VO2 80 nm

--

84,91

--

79,66

89,61

BFO-VO2 30 nm

3,25

75,98

20,77

52,24

--

BFO-VO2 60 nm

1,92

72,64

25,44

42,26

--

9) The major comment concerns the aim of the study, authors should clarify it. Some additional experiments are demanded to support the author’s ideas (ellipsometry). The XPS study should be re-written in the classic way. The comparison of films under study with the systems synthesized for similar purpose is necessary

Answer: Thanks for your comments, the objective of the study has been clarified in the article: “In this article we present a new BFO/VO2/Al2O3 heterostructure with very interesting and promising properties, an analysis of surface, growth rate, roughness and grain size are carried out, in addition, an electrical study was carried out where notable changes in the electrical properties of bilayers when subjected to different temperatures”. We apologize since we do not have ellipsomemetry measurements, however, we corroborate the profilometry measurements by SEM with cross section.

VO2 monolayer cross-sectional view by SEM

 BFO/VO2 bilayer cross-sectional view by SEM

Round 2

Reviewer 2 Report

The manuscript titled “SURFACE AND ELECTRICAL CHARACTERIZATION OF BILAYERS BASED ON BiFeO3 AND VO2” presented by Jhonatan Martínez, Edgar Mosquera-Vargas, Victor Fuenzalida, Marcos Flores, Gilberto Bolaños, Jesús Diosa was presented after revision. I have noted that authors did a good work under the manuscript and answered/commented some questions. However, there are minor issues:

1. R1. “2) It does not clear which samples are described in the sentence “The heterostructures have roughnesses between 0.2 and 16 nm and a grain size between 20 nm and 67 nm.”

Answer: You are right, we have corrected the paragraph clarifying the idea.

The heterostructures, monolayer and bilayer based on BiFeO3 and VO2 grew with good adhesion and without delamination or signs of incompatibility between the layers, have roughness between 0.2 and 16 nm and a grain size between 20 nm and 67 nm.”

It is still unclear; the roughness varies in the wide range for all monolayer and bilayer based on BiFeO3 and VO2. If the roughness and grain size are the major parameters of samples. please, specify the range of variation for each monolayer and bilayer based on BiFeO3 and VO2 samples.

2. It is clear that the authors are not specialists in XPS, but they should give the manuscript for evaluation of XPS part to specialist. It allows authors to avoid some incorrectness.

The chemical composition was studied by means of chemical binding energy of the elements through X-ray photoelectron spectroscopy technique (XPS), using an XPS–Auger PerkinElmer spectrometer model PHI 1257 which includes an ultra-high vacuum chamber, a hemispheric electron energy analyzer and an X-ray source, with Kα radiation unfiltered from an Al (hγ = 1486.6 eV) anode, a step of 0.5 eV.” Should be re-written as “The chemical composition was studied by X-ray photoelectron spectroscopy (XPS), using PHI 1257 spectrometer (PerkinElmer) equipped with a hemispherical electron energy analyzer and an X-ray source with Al Kα radiation (hν = 1486.6 eV).” Some comments to previous version of manuscript, the iris of 0.8 m is too large, it looks that iris was about 8 mm. The sample size is 5x5 mm2 is not a challenge for XPS, the challenge starts with 0.5x0.5 mm2. It looks that that position adjustment was incorrect, or measurement mode did not suitable. Please, for future do not hesitate to take more information about XPS.

The comment concerns the more qualitive presentation of manuscript. Hope, authors consider my recommendations. The manuscript could be accepted after revision.

Author Response

Reviewer #2.

The manuscript titled “SURFACE AND ELECTRICAL CHARACTERIZATION OF BILAYERS BASED ON BiFeO3 AND VO2” presented by Jhonatan Martínez, Edgar Mosquera-Vargas, Victor Fuenzalida, Marcos Flores, Gilberto Bolaños, Jesús Diosa was presented after revision. I have noted that authors did a good work under the manuscript and answered/commented some questions. However, there are minor issues:

  1. R1. “2) It does not clear which samples are described in the sentence “The heterostructures have roughnesses between 0.2 and 16 nm and a grain size between 20 nm and 67 nm.”

Answer: You are right, we have corrected the paragraph clarifying the idea.

The heterostructures, monolayer and bilayer based on BiFeO3 and VO2 grew with good adhesion and without delamination or signs of incompatibility between the layers, have roughness between 0.2 and 16 nm and a grain size between 20 nm and 67 nm.”

It is still unclear; the roughness varies in the wide range for all monolayer and bilayer based on BiFeO3 and VO2. If the roughness and grain size are the major parameters of samples. please, specify the range of variation for each monolayer and bilayer based on BiFeO3 and VO2 samples.

Answer: Thanks for the comment, the changes were made to the manuscript and highlighted in cyan color.

The heterostructures, monolayer and bilayer based on BiFeO3 and VO2 grew with good adhesion and without delamination or signs of incompatibility between the layers. It was observed a good granular arrangement and RMS roughness between 1 and 5 nm for the individual layers (VO2 and BiFeO3) and between 6 and 18 nm for the bilayers (BiFeO3/VO2). Their grain size is between 20 nm and 26 nm for individual layers and between 63 nm and 67 nm for bilayers.

  1. It is clear that the authors are not specialists in XPS, but they should give the manuscript for evaluation of XPS part to specialist. It allows authors to avoid some incorrectness.

“The chemical composition was studied by means of chemical binding energy of the elements through X-ray photoelectron spectroscopy technique (XPS), using an XPS–Auger PerkinElmer spectrometer model PHI 1257 which includes an ultra-high vacuum chamber, a hemispheric electron energy analyzer and an X-ray source, with Kα radiation unfiltered from an Al (hγ = 1486.6 eV) anode, a step of 0.5 eV.” Should be re-written as “The chemical composition was studied by X-ray photoelectron spectroscopy (XPS), using PHI 1257 spectrometer (PerkinElmer) equipped with a hemispherical electron energy analyzer and an X-ray source with Al Kα radiation (hν = 1486.6 eV).” Some comments to previous version of manuscript, the iris of 0.8 m is too large, it looks that iris was about 8 mm. The sample size is 5x5 mm2 is not a challenge for XPS, the challenge starts with 0.5x0.5 mm2. It looks that that position adjustment was incorrect, or measurement mode did not suitable. Please, for future do not hesitate to take more information about XPS.

Answer: Thanks for the comment, the changes were made to the manuscript and highlighted in cyan color.

The comment concerns the more qualitive presentation of manuscript. Hope, authors consider my recommendations. The manuscript could be accepted after revision.
